# LSTM-based-AUTO-Bi-LSTM for Remaining Useful Life (RUL) prediction: the first round of test results

## Abstract

The Remaining Useful Life (RUL) is one of the most critical indicators to detect a component's failure before it effectively occurs. It can be predicted by historical data or direct data extraction by adopting model-based, data-driven, or hybrid methodologies. Data-driven methods have mainly used Machine Learning (ML) approaches, despite several studies still pointing out different challenges in this sense. For instance, traditional ML methods cannot extract features directly from time series depending, in some cases, on the prior knowledge of the system. In this context, this work proposes a DL-based approach called LSTM-based-AUTO-Bi-LSTM. It ensembles an LSTM-based autoencoder to automatically perform feature engineering (instead of manually) with Bidirectional Long Short-Term Memory (Bi-LSTM) to predict RUL. We have tested the model using the Turbofan Engine Degradation Simulation Dataset (FD001), an open dataset. It was generated from the Commercial Modular Aero-Propulsion System Simulation (C-MAPSS) from the Prognostics Center of Excellence (PcoE), from the National Aeronautics and Space Administration (NASA). The objective is to release the first round of analytical results and statistical visualisations of the model application, which will guide us in future improvements.

## 1 Introduction

Cyber-Physical Systems (CPS), Internet of Things (IoT), Internet of Services (IoS), and Data Analytics have built Industry 4.0, which has improved manufacturing efficiency and helped industries face challenges such as economic, social, and environmental (Ruiz-Sarmiento et al., 2020). Condition-Based Maintenance (CBM) performs machines and components' maintenance routines based on their needs, and Prognostics and Health Management (PHM) monitors components' wear evolution using indicators. PHM is a proactive way of implementing CBM by predicting the Remaining Useful Life (RUL), one of the most critical indicators to detect a component's failure before it effectively occurs (Wang et al., 2021; Huang et al., 2019; Wu et al., 2017; Kan et al., 2015).

RUL can be predicted by historical data or direct data extraction by adopting model-based, data-driven, or hybrid methodologies. Model-based methods are challenging, expensive, and time-consuming to develop in complex equipment due to the need for prior system knowledge. Data-driven methods have mainly used Machine Learning (ML) approaches. They are less complex and expensive, more applicable and provide a suitable trade-off between complexity, cost, precision, and applicability (Cheng et al., 2021; Mrugalska, 2019; Li et al., 2019; Yang et al., 2016), although they require large amounts of historical data for development (Liewald et al., 2022)

Meanwhile, despite the increased use of ML to predict RUL, several studies have still pointed out different challenges in this sense (Huang et al., 2019). For example, most ML methods' accuracy in predicting RUL largely depends on the feature extraction quality, and their performance is affected in the case of very complex systems with multiple components, multiple states, and a considerable amount of parameters (Zhao et al., 2021; Chen et al., 2019). Moreover, the literature has also reported that most of these models do not consider operation conditions; the machines operate in different states, even on the same shop floor. It significantly impacts the degradation behaviour and raw sensor signals that may be non-stationary, nonlinear, and mixed with much noise (Liu et al.,

2020a). Finally, traditional ML methods cannot extract features directly from time series depending on the complex intermediate transformation and, in some cases, depending on the prior knowledge of the system (Cabrera et al., 2020).

To overcome several challenges and improve the accuracy of RUL prediction, there has been a prominent use of Deep Learning (DL) Methods, especially Recurrent Neural Networks (RNN) and Long Short-Term Memory (LSTM), besides other variations (Zhu et al., 2019; Li et al., 2020; Liu et al., 2020b). They have emerged and achieved outstanding results in different areas due to their strong capacity to map the relationship between degradation paths and measured data. Also, these methods can learn feature representation automatically, such that it is not necessary to design features manually, eliminating the need for previous knowledge of the system (Zhu et al., 2019). Finally, DL methods have a high capacity to deal with many complex data (Kong et al., 2019) [17]. Nonetheless, the literature reports some drawbacks, such as the data deficit issue, especially considering the varying operation conditions and the degradation mode of the components in practical industrial applications (Liu et al., 2020a).

In this context, Ferreira & Gonçalves (2022),among other results, have mapped 14 challenges in using ML methods for RUL prediction and pointed out some approaches used in the literature to overcome these challenges. From this collection of approaches, it was possible to propose an architecture called LSTM-based-AUTO-Bi-LSTM, which ensembles an Autoencoder (Unsupervised/Reconstructive Learning Technique) with the DL method Bidirectional Long Short-Term Memory (Bi-LSTM). The autoencoder aims to perform feature engineering automatically (instead of manually). The Bi-LSTM aims to predict the RUL based on the outputs of the autoencoder. This type of ensembling is, at least, very few applied in the RUL prediction process. To test our model, we have explored the turbofan engine problem through the dataset gathered from PCoE/NASA. Therefore, this work aims to release the first round of analytical results and statistical visualisations of the model application.

The remaining of this work is as follows. Section 2 describes the problem and the used dataset, and Section 3 introduces the LSTM-based-AUTO-Bi-LSTM architecture. Section 4 describes the experimental context, and Section 5 presents the results and compares them with the literature. Finally, Section 6 concludes this work by giving some directions for future works.

## 2 THE PROBLEM AND DATASET

### 2.1 THE PROBLEM

PHM has been an essential topic in the industry for predicting the state of assets to avoid downtime and failures (NASA, 2022). In the aircraft industry, attempted maintenance is critical to ensure operation safety (Zheng et al., 2018), besides increasing economic efficiency (Deng et al., 2019). According to the International Air Transport Association (IATA), maintenance costs of the major aviation companies reached $15.57 billion between 2012 and 2016, which represented a growth of 3% (Kraus & Feuerriegel, 2019). Turbofan engines, specifically, are responsible for about 30% of the failures in an aircraft, and in great-proportion accidents, these systems have been the root cause in 40% of the cases. Besides, propulsion device maintenance costs share about 40% of the full aircraft maintenance costs (Tang et al., 2021). The main components of a turbofan engine include the fan, low-pressure compressor (LPL), high-pressure compressor (HPC), combustor, high-pressure turbine (HPT), and low-pressure turbine (LPT), and nozzle.

### 2.2 THE DATASET

The dataset was gathered from the Prognostics Center of Excellence – PCoE, from the National Aeronautics and Aerospace Administration (NASA). In this sense, the information provided in this subsection was retrieved from that source NASA (2022) and Saxena et al. (2008).

Engine degradation simulation was carried out using Commercial Modular Aero-Propulsion System Simulation (C-MAPSS). Four different datasets (FD001, FD002, FD003, and FD004) were simulated under various operational conditions. They comprised a range of values for three operating conditions – Altitude, from 0 to 42K ft., Mach Number, from 0 to 0.84, and Throttle Resolver Angle (TRA), from 20 to 100 – and fault modes – High-Pressure Compressor Degradation or/and Fan

Degradation – combinations. Records of several sensor channels to characterise fault evolution. The objective of these datasets is to predict the RUL of each engine in the test dataset. RUL can be defined as the equivalent number of flights remaining for the engine after the last data point in the test dataset.

The datasets consist of multiple multivariate time series. Each data set is further divided into training (train_FD001, train_FD002, train_FD003, and train_FD004) and test subsets (test_FD001, test_FD002, test_FD003, and test_FD001). Each time series is from a different engine, i.e., the data can be considered from a fleet of engines of the same type. Each engine starts with different degrees of initial wear and manufacturing variation, unknown to the user. This wear and variation are considered normal, i.e., it is not considered a fault condition.

Three operational settings substantially affect engine performance were also included in the datasets: Altitude, Mach Number, and TRA. The data is contaminated with sensor noise. The engine usually operates at the start of each time series, developing a fault at some point. In the training dataset, the fault grows in magnitude until system failure. The time series sometimes ends before system failure in the test dataset. It also provides a vector of true Remaining Useful Life (RUL) values for the test data (RUL_FD001, RUL_FD002, RUL_FD003, and RUL_FD001). The train and test datasets are presented through 26 columns of numbers (Unit/Engine Number, Time (Cycles), Operational Setting 1, . . . , Operational Setting 3, Sensor Measurement 1, . . . , Sensor Measurement 21), comprehending a different variable. Each row is a snapshot of data taken during a single operational cycle. Table A1 presents a summarised description of the datasets, and Table A2 describes the sensor measurement variables (columns 6 to 26).

## 3 THE LSTM-BASED-AUTO-BI-LSTM

The proposed architecture consists of ensembling two methods. First, we used an LSTM-based Autoencoder to perform automatic feature engineering (instead of manually) through the raw dataset. Then we applied Bi-LSTM initialised through the autoencoder outputs to predict the RUL (prediction model). Both methods are explained in detail, and an overview of the entire architecture is presented in the following subsections.

### 3.1 LSTM-BASED AUTOENCODER

The autoencoders are unsupervised NN structures (Ren et al., 2021), completely symmetrical (Ren et al., 2018), with one input layer, a hidden layer, and one output layer (Xia et al., 2019). The input layer and the first half of the hidden layer build the encoder. The second half of the hidden layer and the output layer build the decoder. The fundamental objective is reconstructing original data by minimising the error between the network output data and the original data and initialising a deep NN (Xia et al., 2019). The number of nodes in each hidden layer is less than the number of nodes in the input layer and the output layer, creating a type of bottleneck (Chen et al., 2020). These structures have improved the optimal model determination through the random network initialisation (Xia et al., 2019) and have also demonstrated great ability for feature extraction (Chen et al., 2020). The generic mathematical autoencoder construction is shown in equations (1) and (2).

$$y = \sigma(Wx + b) \tag{1}$$
$$x' = \sigma[W'y + b] \tag{2}$$

In equation (1), y represents the features learned by the encoder (code), x represents the input vector, W is the weight matrix between the input layer and the hidden layer, b is the bias, and $\sigma$ is the activation function. In equation (2), x' represents the vector constructed through the features learned from the hidden layer, and W' is the weight matrix between the input layer and the hidden layer. The remaining parameters are the same in equation (1).

In this work, we have set up an autoencoder by applying Long Short-Term Memory (LSTM) model in the encoder and decoder layers (LSTM-based Autoencoder). The main idea behind the LSTM is to capture the dependence of the current state on the previous state, which means in the forward direction (Zhao et al., 2017). The general LSTM cell (neuron) structure can be divided into three parts or gates. First, the input gate decides when the model can receive a new information state. Next,

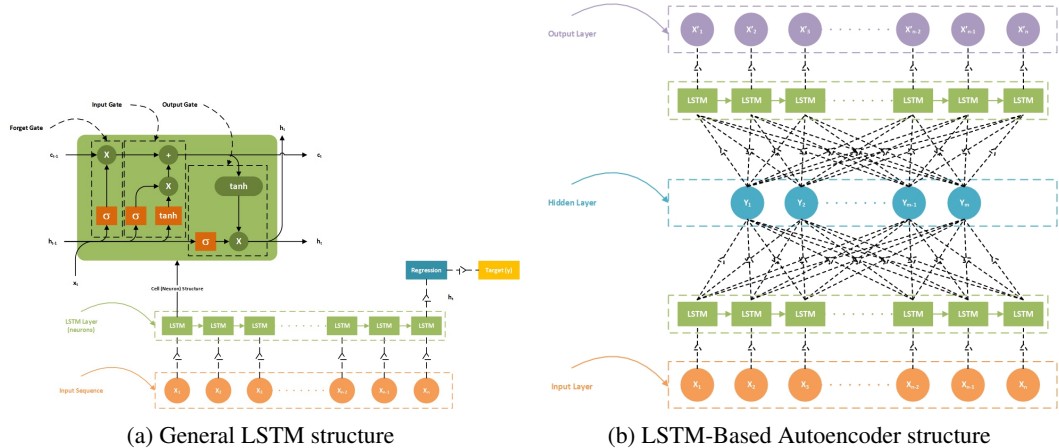

(a) General LSTM structure         (b) LSTM-Based Autoencoder structure

Figure 1: General LSTM and LSTM-Based-Autoencoder structure.

the forget gate decides when the model can forget the previous information state. Finally, the output gate decides which information to output. These gates are determined by the input at the current instant and the output at the previous instant through sigmoid and hyperbolic tangent activation functions (Liu et al., 2021b). Figure 1 presents the general LSTM structure and the LSTM-Based Autoencoder structures.

## 3.2 THE BI-LSTM

The Bi-LSTM derives from the LSTM, adapted to learning the most representative features hidden in the Condition Monitoring (CM) data (Huang et al., 2019). With Bi-LSTM, the main idea is to capture past and future information by processing the CM data forward and backwards through two separate hidden layers. Equations sets (3) and (4) describe the separated hidden layers function at step t. The symbols $\rightarrow$ and $\leftarrow$ denote, respectively, the forward and backward process (Huang et al., 2019). Deeper detailed considerations regarding Bi-LSTM can be found in Zhao et al. (2017) and Zhang et al. (2018a). Figure 2 shows the general Bi-LSTM structure, and Figure 3 shows an overview of the proposed architecture.

$$
\overrightarrow{h}_t = f(\overrightarrow{x}_t, \overrightarrow{h}_{t-1}; \overrightarrow{\Theta}_{LSTM}) =
$$

$$
\begin{cases}
\overrightarrow{z}_t = \tanh(\overrightarrow{W}_z \overrightarrow{x}_t + \overrightarrow{R}_z \overrightarrow{h}_{t-1} + \overrightarrow{b}_z) \\
\overrightarrow{i}_t = \sigma(\overrightarrow{W}_i \overrightarrow{x}_t + \overrightarrow{R}_i \overrightarrow{h}_{t-1} + \overrightarrow{b}_i) \\
\overrightarrow{f}_t = \sigma(\overrightarrow{W}_f \overrightarrow{x}_t + \overrightarrow{R}_f \overrightarrow{h}_{t-1} + \overrightarrow{b}_f) \\
\overrightarrow{o}_t = \sigma(\overrightarrow{W}_o \overrightarrow{x}_t + \overrightarrow{R}_o \overrightarrow{h}_{t-1} + \overrightarrow{b}_o) \\
\overrightarrow{c}_t = \overrightarrow{z}_t \odot \overrightarrow{i}_t + \overrightarrow{c}_{t-1} \odot \overrightarrow{f}_t \\
\overrightarrow{h}_t = \tanh(\overrightarrow{c}_t) \odot \overrightarrow{o}_t
\end{cases}
\tag{3}
$$

$$
\overleftarrow{h}_t = f(\overleftarrow{x}_t, \overleftarrow{h}_{t-1}; \overleftarrow{\Theta}_{LSTM}) =
$$

$$
\begin{cases}
\overleftarrow{z}_t = \tanh(\overleftarrow{W}_z \overleftarrow{x}_t + \overleftarrow{R}_z \overleftarrow{h}_{t-1} + \overleftarrow{b}_z) \\
\overleftarrow{i}_t = \sigma(\overleftarrow{W}_i \overleftarrow{x}_t + \overleftarrow{R}_i \overleftarrow{h}_{t-1} + \overleftarrow{b}_i) \\
\overleftarrow{f}_t = \sigma(\overleftarrow{W}_f \overleftarrow{x}_t + \overleftarrow{R}_f \overleftarrow{h}_{t-1} + \overleftarrow{b}_f) \\
\overleftarrow{o}_t = \sigma(\overleftarrow{W}_o \overleftarrow{x}_t + \overleftarrow{R}_o \overleftarrow{h}_{t-1} + \overleftarrow{b}_o) \\
\overleftarrow{c}_t = \overleftarrow{z}_t \odot \overleftarrow{i}_t + \overleftarrow{c}_{t-1} \odot \overleftarrow{f}_t \\
\overleftarrow{h}_t = \tanh(\overleftarrow{c}_t) \odot \overleftarrow{o}_t
\end{cases}
\tag{4}
$$

In the equations sets (3) and (4), $\overrightarrow{\Theta}_{LSTM}$ and $\overleftarrow{\Theta}_{LSTM}$ are the parameters set of the forward and backward processes, shared by all the time steps and learned during model training. $\overrightarrow{W}_k, \overleftarrow{W}_k \in$

$\mathbb{R}^{L \times p}$ are input weights (related to $\overrightarrow{x}_t, \overleftarrow{x}_t$) of the forward and backward process, respectively. $\overrightarrow{\mathbb{R}}_k, \overleftarrow{\mathbb{R}}_k \in \mathbb{R}^{L \times L}$ are recurrent weights (related to $\overrightarrow{h}_{t-1}, \overleftarrow{h}_{t+1}$) of the forward and backward process, respectively. $\overrightarrow{b}_k, \overleftarrow{b}_k \in \mathbb{R}^L$ are bias weights of the forward and backward process, respectively. Finally, $\sigma$ (logistics sigmoid) and $\tanh$ (hyperbolic tangent) are pointwise nonlinear activation functions, $\odot$ denotes pointwise multiplication of two vectors, L denotes de dimensionality of the hidden neurons and $k \in \{Z, i, f, o\}$.

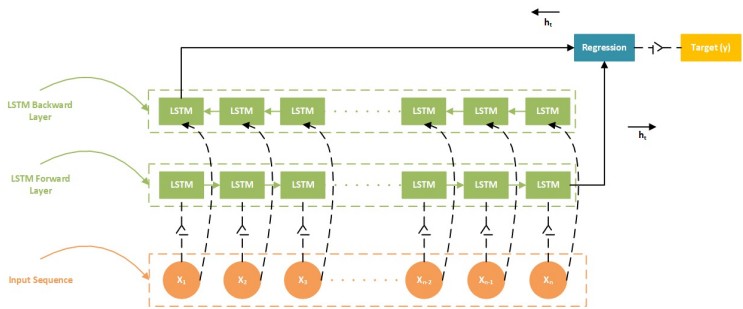

Figure 2: Bi-LSTM structure.

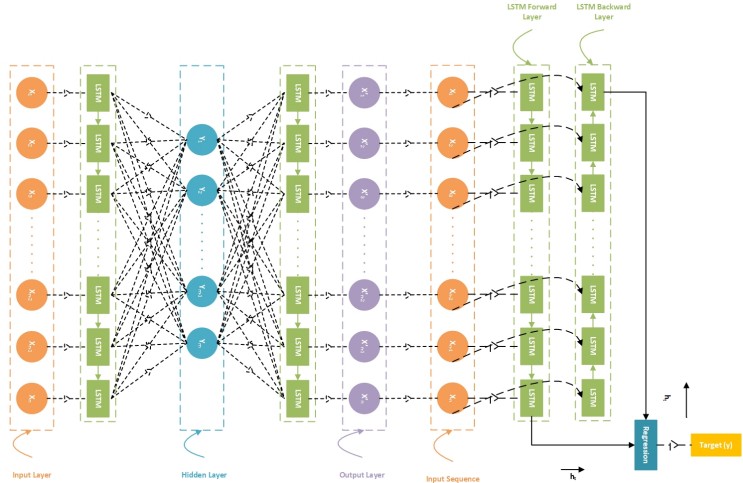

Figure 3: LSTM-based-AUTO-Bi-LSTM.

## 4 EXPERIMENTAL CONTEXT

### 4.1 NORMALISATION

The normalisation was performed by applying the z-score method, as described in Che et al. (2019); Wang et al. (2019); Jiang et al. (2020). The mean of the normalised values is 0, and the standard deviation of the normalised values is 1. The normalised values represent the number of standard deviations that the original value is from the mean and are calculated by the equation (5).

$$z_i = \frac{(x_i - \mu_i)}{\sigma_i} \tag{5}$$

The $z_i$ is the normalised value; $x_i$ is the raw data in the sensor i; $\mu_i$ is the average value of the *ith* sensor, and $\sigma_i$ is the standard deviation of the *ith* sensor.

## 4.2 PIECE-WISE FUNCTION

In real applications, a machine component degrades less at the beginning of life. On the other hand, the degradation increases as it is close to its end of life (Zheng et al., 2017). It means that the component degradation is early unclear, and the RUL of similar sensor data may vary sensible (Saxena et al., 2008). For instance, Figure 4 shows ten selected sensors from engine 65 of the training dataset. As we can perceive, for all chosen sensors, it is only possible to obtain some trend, approximately from cycle 68. A Piece-Wise linear RUL target function was assumed. The maximum threshold for RUL was 125 to deal with this condition and better model the RUL behaviour throughout time, such as in several literature examples (see, among others, Mrugalska (2019); Liu et al. (2020b); Saxena et al. (2008); Jiang et al. (2020)). This is a crucial concern since if a large RUL is assumed, there can be a significant fluctuation of the predicted RUL in the early stage. Otherwise, if a small RUL is considered, the predicted RUL can be confined to a small range (Saxena et al., 2008).

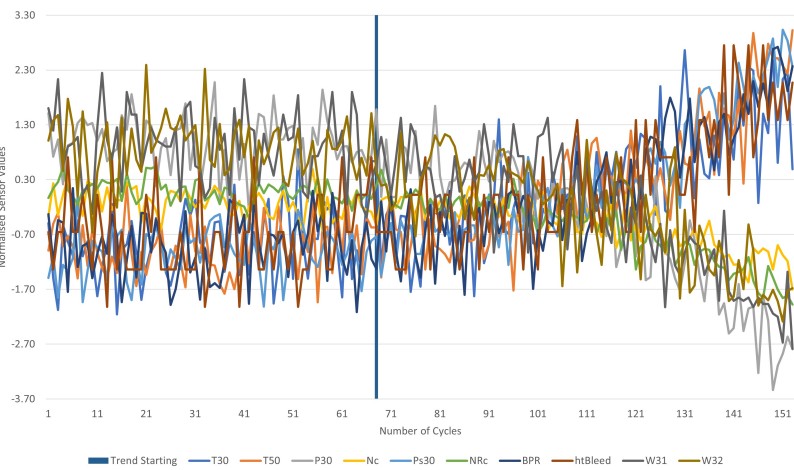

Figure 4: Selected sensors of engine 65 from the FD001 training dataset.

## 4.3 PERFORMANCE EVALUATION CRITERIA

To evaluate the proposed method's performance, we have adopted the Root Mean Square Error (RMSE) between the predicted RUL and the ground truth RUL, which is a standard metric reported in the literature (see, among other, Li et al. (2019); Zheng et al. (2018); Falcon et al. (2020); Zhao et al. (2019)). It allowed us to compare our results with the literature state-of-the-art mathematically. The RMSE gives equal weights for early and late predictions and is defined by equation (6).

$$RMSE = \sqrt{\frac{1}{N}\sum_{i=1}^{N}(\widehat{RUL}_i - RUL_i)^2} \tag{6}$$

In equation (6), RMSE is the computed error; N is the total number of data samples, and $\widehat{RUL}_i$ and $RUL_i$ are respectively the predicted RUL and the ground truth RUL concerning the *ith* data point.

## 4.4 LSTM-BASED-AUTO-BI-LSTM SETUP

The proposed architecture was tested in two ways. First, an utterly aleatory setup was performed regarding the hyperparameters. This setup was considered the initial setup or baseline setup. Second, the literature was surveyed more in-depth, and some hyperparameters were changed accordingly. Specifically, regarding the number of epochs in the LSTM-based Autoencoder, we have performed specific runs to determine the best number. This setup was considered the testing setup. The initial setup and testing setup are fully detailed as follows.

### 4.4.1 Initial setup (baseline setup)

- **LSTM-based Autoencoder** ⇒ LSTM-based layers (encoder and decoder) with 24 neurons each (number of features in the dataset). Optimizer = ADAM and Loss Function = Binary Crossentropy. Validation Split = 10% (standard throughout the literature). Epochs = 32. Batch-size = 50 (about 0.25% of the training dataset). Time-Window = 31 (minimum engine running length in the test dataset).
- **Bi-LSTM** ⇒ Single layer with Activation Function = Hyperbolic Tangent. Number of Neurons = 100. Selected features (automatic selection) from the LSTM-based Autoencoder = 12. Dense = 30, with Activation Function = ReLu. Output Layer with Activation Function = Linear. Optimizer = RMSProp and Loss Function = MSE. Epochs = 32. Batch-size = 200 (about 1% of the training dataset). Time-Window = 31 (minimum engine running length in the test dataset).

### 4.4.2 Testing setup

- **LSTM-based Autoencoder** ⇒ LSTM-based layers (encoder and decoder) with 24 neurons each (number of features in the dataset). Optimizer = ADAM and Loss Function = MSE (most common used). Validation Split = 10% (standard throughout the literature). Epochs = 50 (see next section). Batch-size = 50 (about 0.25% of the training dataset). Time-Window = 31 (minimum engine running length in the test dataset).
- **Bi-LSTM** ⇒ Single layer with Activation Function = Hyperbolic Tangent. Number of Neurons in the set {24, 48, 72, 96, 120, 144, 168, 192, 216, 240}. Selected features (automatic selection) from the LSTM-based Autoencoder in the set {6, 12, 18, 24}. Dense = 30, with Activation Function = ReLu (most common used). Output Layer with Activation Function = Linear (most common used). Optimizer = ADAM and Loss Function = MSE (most common used). Epochs = 32. Batch-size = 200 (about 1% of the training dataset). Time-Window = 31 (minimum engine running length in the test dataset).

In this work, four different setups were tested for the LSTM-based Autoencoder, and 41 different configurations (one for baseline and 40 for testing setup) were tested for the LSTM-based-AUTO-Bi-LSTM. The results and comparison with state of the art in literature are presented in the next Section. To reduce the influence of random factors, the reported results in the next Section are the average of ten independent runs. Finally, all the experiments were performed using Colaboratory from Google Research (Colab).

## 5 Results

First, we have analysed the effective gain after each epoch in the LSTM-based Autoencoder, considering the loss (training and validation) and accuracy (training and validation). The objective was to determine the best value for this hyperparameter. The analysis also considered the processing time for 32, 64, 96 and 128 epochs. Regardless of the number of epochs setup, we could observe that, from epoch 32, the expressive gain in loss and accuracy decreases fast. However, the processing time of the LSTM-based Autoencoder increases as quickly as we increase the number of epochs (329.54, 628.79, 928.83 and 1169.35 seconds, respectively). Thus, considering this trade-off (processing time against effective gain) and aiming to avoid possible discrepancies (due to the stochastic nature of the method) the best value considered was 50 epochs. Figure 5 shows the loss (a) and accuracy (b) for the LSTM-based autoencoder.

Second, we have analysed the behaviour of the Bi-LSTM by fixing the LSTM-based Autoencoder setup and varying the configuration of the number of neurons and selected features. The remaining hyperparameters were as in the testing setup. The objective was to compare the results with our baseline and the literature's state of the art. To do this, we ran the LSTM-based-AUTO-Bi-LSTM 400 times (about 106 running hours) through 40 different configurations. The results are graphically presented as follows. Figure 6 shows the RMSE evolution throughout the tests performed. The three best averages are marked in the graph, with two occurring using 24 features and one using only six features. It is also possible to perceive that the lines for 12, 18 and 24 features have a smoother path than the line for six features. Also, the final trend appears to decrease (faster with 24 features) in those lines while it seems to increase with six features.

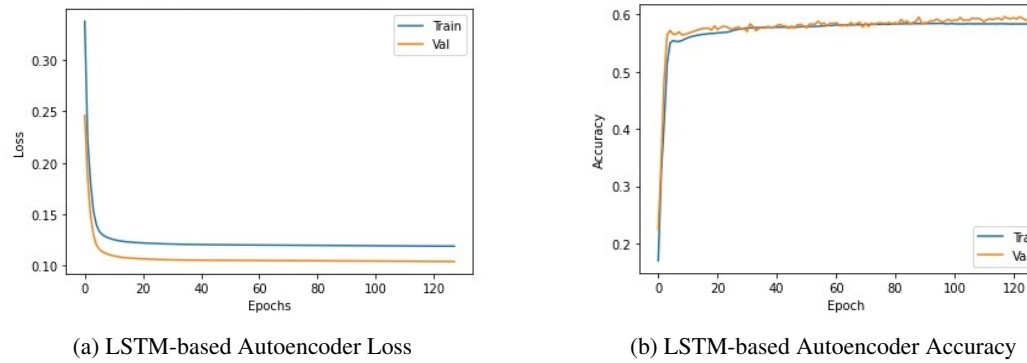

(a) LSTM-based Autoencoder Loss

(b) LSTM-based Autoencoder Accuracy

Figure 5: LSTM-Based-Autoencoder Loss and Accuracy

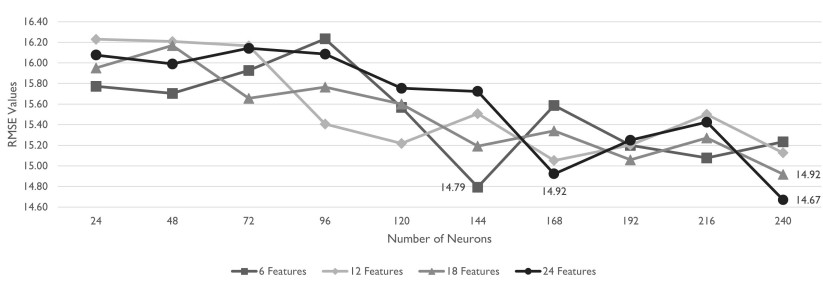

Figure 6: RMSE evolution throughout the tests (average of ten runs).

Still, regarding the behaviour of the Bi-LSTM, we have plotted the boxplots of the tests performed considering 6 (a), 12 (b), 18 (c) and 24 (d) features. Figure 7 shows these results, and it is possible to visualise that the more stable distribution relies on 18 features, which also contain the lowest number of outliers.

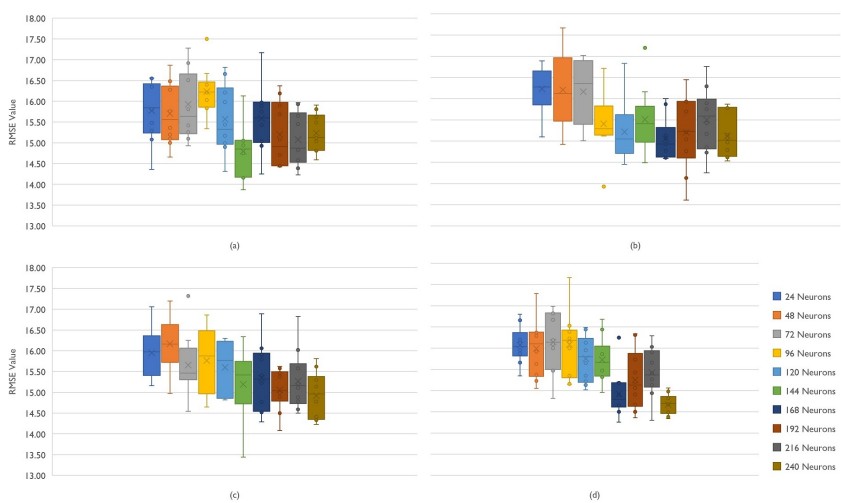

Figure 7: Distributions of the tests performed considering 6, 12, 18 and 24 features.

Third, we have captured the total average processing time of the LSTM-based-AUTO-Bi-LSTM. Figure 8 shows the evolution of this time. The minimum time obtained was 471.38 seconds (using 24 features in the Bi-LSTM), and the maximum was 1768.08 seconds (using 18 features in the Bi-LSTM). Also, it is possible to observe a more stable trend when using 24 features in the Bi-LSTM.

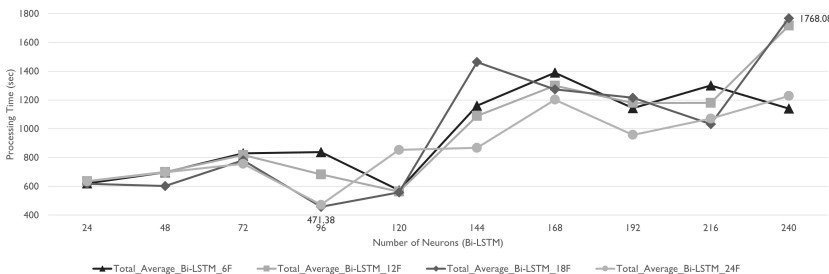

Figure 8: LSTM-based-AUTO-Bi-LSTM total average processing time.

Finally, we compared our results with the state-of-the-art literature. Figure 9 highlights the RMSE average of the baseline setup (17.38 in red) and the best RMSE average of the testing setup (14.67 in blue) with the state-of-the-art literature. Considering the testing setup result, it is possible to see that it is well positioned in terms of the studied dataset. Indeed, from the architectures designed for automatic feature extraction (dark grey), only one (Bi-LSTM + CNN) outperforms the result in terms of RMSE. The remaining architectures that outperform the proposed one must perform some feature (or sensor) selection. This implies previous knowledge regarding the system and makes it challenging to apply them directly. A list of the methods and their acronyms is presented in Table A3. Additional characteristics of the state-of-the-art methods are shown in Table A4. The symbol N.A. means that data/information did not apply to the method or is not available in the text to the best of our evaluation.

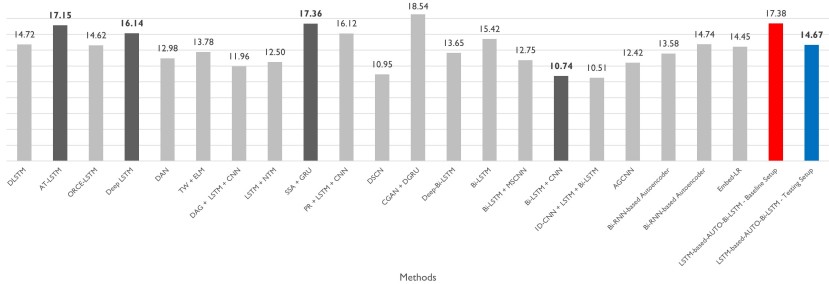

Figure 9: State-of-the-art Results (RMSE - Dataset FD001).

## 6  CONCLUSIONS

The LSTM-based-AUTO-Bi-LSTM architecture was shown in these initial tests to be within the state-of-the-art literature for subdataset FD001, considering the results demonstrated. Especially when compared with methods that performed automatic feature extraction, our architecture proved far superior to the others, except for the work in Remadna et al. (2020). Although Colab may have presented some instability and influenced the capture of processing times, it is estimated that the obtained times are reasonable regarding the number of instances used.

First, we want to vary the number of neurons in the encoder and decoder layers of the LSTM-based autoencoder. Then, we want to test different hyperparameters from those tested in this work, namely varying the number of epochs used and the number of layers used in the predictive model (Bi-LSTM). Next, we intend to perform more tests using the subdatasets FD002, FD003 and FD004, which are more complex in terms of operational conditions and fault modes. Further, we intend to evaluate these future results using the Score Function introduced in Saxena et al. (2008) jointly with RMSE. Finally, we will test this architecture with categorical datasets generated through embedded monitoring systems (system log files).

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

## A    APPENDIX

Table A1: C-MAPSS Dataset Description

| Description | C-MAPSS Dataset | | | |
| --- | --- | --- | --- | --- |
| | FD001 | FD002 | FD003 | FD004 |
| Training Engines Number | 100 | 260 | 100 | 248 |
| Testing Engines Number | 100 | 259 | 100 | 248 |
| Operational Conditions | 1 | 6 | 1 | 6 |
| Fault Modes | 1 | 1 | 2 | 2 |
| Max. Life Spam (Cycles) | 362 | 378 | 525 | 546 |
| Min. Life Spam (Cycles) | 128 | 128 | 145 | 128 |
| Train Instances (rows) | 20.631 | 53.759 | 24.720 | 61.249 |
| Test instances (rows) | 13.096 | 33.991 | 16.569 | 41.214 |

Table A2: Sensor measurement variables description

| Variable (Sensor Measurement) | Description | Unit |
|---|---|---|
| T2 | The total temperature at the fan inlet | °R |
| T24 | The total temperature at the LPC outlet | °R |
| T30 | The total temperature at the HPC outlet | °R |
| T50 | The total temperature at the LPT outlet | °R |
| P2 | Pressure at fan inlet | psia |
| P15 | Total pressure in bypass-duct | psia |
| P30 | Total pressure at the HPC outlet | psia |
| Nf | Physical fan speed | Rpm |
| Nc | Physical core speed | rpm |
| epr | Engine pressure ratio (P50/P2) | — |
| Ps30 | Static pressure at HPC outlet | psia |
| phi | Ratio of fuel flow to Ps30 | pps/psi |
| NRf | Corrected fan speed | pm |
| NRc | Corrected core speed | rpm |
| BPR | Bypass Ratio | — |
| farB | Burner fuel-air ratio | — |
| htBleed | Bleed Enthalpy | — |
| Nf_dmd | Demanded fan speed | rpm |
| PCNfR_dmd | Demanded corrected fan speed | rpm |
| W31 | HPT coolant bleed | lbm/s |
| W32 | LPT coolant bleed | lbm/s |

Table A3: Methods described in the literature and their acronyms

| Method | Acronym |
|---|---|
| Anomaly Triggered Long Short-Term Memory | AT-LSTM |
| Bidirectional Gated Recurrent Units | BGRU |
| Bidirectional Long Short-Term Memory | Bi-LSTM |
| Deep- Bidirectional Long Short-Term Memory | Deep-Bi-LSTM |
| Conditional Generative Adversarial Network | CGAN |
| Convolutional Neural Networks | CNN |
| Deep & Attention Network | DAN |
| Deep Gated Recurrent Unit Network | DGRU |
| Deep Long Short-Term Memory | Deep LSTM |
| Deep Separable Convolutional Network | DSCN |
| Direct Acyclic Graphic Network | DAG |
| D-Long Short-Term Memory | DLSTM |
| Extreme Learning Machine | ELM |
| Feature Attention Mechanism BRGU CNN | AGCNN Gated Recurrent Unit |
| GRU | |
| Long Short-Term Memory | LSTM |
| Multiscale Convolutional Neural Networks | MSCNN |
| Neural Turing Machine | NTM |
| Ordinal Regression Censored Estimation Long Short-Term Memory | ORCE-LSTM |
| Polynomial Regression | PR |
| Stacked Sparse Autoencoder | SSA |
| Time Window | TW |

Table A4: Additional characteristics of the state-of-the-art methods

| Method | Learning Rate | Epochs | Batch Size | Max RUL | Features Selected | RMSE Calculation | Ref. |
|---|---|---|---|---|---|---|---|
| PR+LSTM+CNN | 0.001 | 500 | 250 | N.A. | N.A. | N.A. | [(Kong et al., 2019)] |
| TW+ELM | N.A. | N.A. | N.A. | 125 | 14 | N.A. | [(Zheng et al., 2018)] |
| SSA+GRU | N.A. | N.A. | N.A. | 125 | Auto. Selection | Aver. in 10 tests | [(Deng et al., 2019)] |
| DSCN | N.A. | N.A. | N.A. | 130 | 14 | Aver. in 20 tests | [(Wang et al., 2019)] |
| Bi-LSTM+MSCNN | 0.00001 | 300 | 64 | 125 | N.A. | N.A. | [(Jiang et al., 2020)] |
| LSTM+NTM | 0.005 | 50 | 100 | 130 | 14 | N.A. | [(Falcon et al., 2020)] |
| DLSTM | N.A. | N.A. | N.A. | 130 | 13 | N.A. | [(Zhao et al., 2019)] |
| AT-LSTM | N.A. | N.A. | N.A. | 125 | Auto. Selection | N.A. | [(Aydemir & Acar, 2020)] |
| ORCE-LSTM | 0.001/ 0.005 | Max2000 | 32 | 130 | N.A. | N.A. | [(TV et al., 2019)] |
| DeepLSTM | N.A. | N.A. | N.A. | 130 | Auto. Selection | N.A. | [(Zheng et al., 2017)] |
| DAN | 0.001 | 500 | 256 | 125 | N.A. | N.A. | [(Liu & Wang, 2021)] |
| CGAN+DGRU | 0.0001/ 0.001 | 100/100 | 54/512 | 130 | N.A. | Aver. in 5 tests | [(Behera & Misra, 2021)] |
| Deep-Bi-LSTM | N.A. | Max300 | N.A. | 125 | 14 | N.A. | [(Wang et al., 2018)] |
| Bi-LSTM | 0.01/ 0.015 | 500/140 | 30 | 130 | 8 | N.A. | [(Zhang et al., 2018b)] |
| Bi-LSTM+CNN | 0.0001 | Max2000 | N.A. | N.A. | Auto. Selection | N.A. | [(Remadna et al., 2020)] |
| DAG+LSTM+CNN | 0.005 | 40 | 200 | 125 | 14 | N.A. | [(Li et al., 2019)] |
| Embed-LR | N.A. | N.A. | N.A. | 120 | N.A. | N.A. | [(Gugulothu et al., 2017)] |
| 1D-CNN + LSTM + Bi-LSTM | 0.0001 | 200 | 200 | N.A. | 14 | N.A. | [(Hong et al., 2020)] |
| AGCNN | N.A. | N.A. | N.A. | N.A. | 14 | N.A. | [(Liu et al., 2021a)] |
| Bi-RNN | 0.005 to 0.05 | N.A. | N.A. | 135 | 14 | N.A. | [(Yu et al., 2020)] |
| Bi-RNN | 0.02 | 2 | N.A. | N.A. | 14 | N.A. | [(Yu et al., 2019)] |

