# OpenReview forum: "LSTM-BASED-AUTO-BI-LSTM for Remaining Useful Life (RUL) Prediction: the first round of test results"
_ICLR.cc/2023/Conference — Submitted to ICLR 2023_

### Official Review · Reviewer_5eu9 · 2022-10-25

**Confidence:** 4
**Correctness:** 2
**Technical Novelty And Significance:** 1
**Empirical Novelty And Significance:** 2
**Recommendation:** 3

**Clarity, Quality, Novelty And Reproducibility:**

* The paper clearly explains the method (especially with the images), but both setups are not trivial to understand.
* No part of the method is entirely new.
* Code is not available.

**Strength And Weaknesses:**

* Strength:
    * Images are nicely made and help explaining the method and how both parts are combined.
    * The authors perform hyperparameter search to help finding a good configuration overall.

* Weaknesses:
    * The method's name is more complex and confusing than it should be.
    * I actually did not understand the general idea of the proposed approach. The rationale behind deep learning models is to learn/extract features automatically, so why not using only a single bi-LSTM to learn how to extract them? There isnt enough support for the autoencoder in the beginning, especially with only 24 features.
    * The initial setup is not clear in defining losses: how is the Binary Cross-entropy used as the autoencoder loss? I think that needs further explanations, since it is not trivial. Also, if the authors claim that MSE is more commonly used, why not use it?
    * In the Bi-LSTM, RMSProp is an optimization approach, not a loss function. Which loss function was used to train it?
    * The method is compared to 16 others, but 10 of them are better than the proposed approach, and the best one is a method that also uses deep learning for feature extraction. No argument is provided to support why the proposed approach is better or has any advantage over the baselines.
    * Figure 5 does not help explaining anything. Processing time and epochs represent the same information (time), so it is not needed to demonstrate both. Also, too much information is shown, I would've selected only train or val gain, not both.
    * It would be good to include the input and output in Figure 1(b).
    * There is no need to explain all four datasets if only one of them is actually used.

**Summary Of The Paper:**

* The authors propose a method for predicting the Remaining Useful Life (RUL) of components. They combine an LSTM-based autoencoder to extract features and a Bi-LSTM network to predict RUL. They validate the approach in one dataset and make an extensive test of hyperparameters and execution setups. In the end, they compare the method with baselines from the literature.

**Summary Of The Review:**

* The paper clearly propose a different approach and uses good images to help explaining it. Unfortunately, many aspects lack justification, and the experimental setups are bad detailed (with possible mistakes). They do not introduce nothing entirely new, and when the methodology is compared with other work it underperforms.

---

### Official Review · Reviewer_fVJc · 2022-10-29

**Confidence:** 4
**Correctness:** 2
**Technical Novelty And Significance:** 2
**Empirical Novelty And Significance:** 2
**Recommendation:** 3

**Clarity, Quality, Novelty And Reproducibility:**

The paper is clear and logically structured. Unfortunately, the code is unavailable, making it difficult to reproduce the experiments.
The idea of using autoencoders and LSTM layers to model time series information is not new. The main concern of this submission is the novelty. The proposed method sounds too incremental from existing works. The scalability of the approach needs to be highlighted. I’m not convinced the method is generic to all problems, and more experiments are needed.


**Details Of Ethics Concerns:**

I have no ethical concerns about the paper.


**Strength And Weaknesses:**

+ The proposed approach is interesting and presents promising results.
+ The idea of using autoencoders and LSTM layers to model time series is not new and the authors should emphasize the novelty of the proposed approach.
- I miss some qualitative discussions. The authors explain this is the first round of test results, but I was expecting more insights and conclusive analysis.
- The authors did not discuss the limitation of the proposed method. Therefore, it will be meaningful to discuss the gap between the experiments in the current version of the paper and the real-world applications.
- It is an interesting paper, but it seems to be an ongoing project yet.

**Summary Of The Paper:**

This paper proposes a method combining autoencoders and Bi-LSTM layers to predict the remaining useful life of physical equipments. The author argues that besides promising approaches, traditional ML methods cannot extract sufficient information from the equipment data and they need manual specialist inputs to obtain consistent results. The idea is to use the proposed approach to extract automatic features from the equipment time series data and predict the remaining useful life information. The method was evaluated in the Turbofan Engine Degradation Simulation Dataset and the preliminary results are presented.

**Summary Of The Review:**

The proposed paper presents a promising approach with interesting results. The analysis is direct, but I was expecting more insights. The authors also should discuss the proposed approach's novelty and limitations. It seems to be an ongoing project yet.

---

### Official Review · Reviewer_Lf7p · 2022-11-03

**Confidence:** 4
**Correctness:** 2
**Technical Novelty And Significance:** 1
**Empirical Novelty And Significance:** 2
**Recommendation:** 3

**Clarity, Quality, Novelty And Reproducibility:**

Clarity:
The paper is decently written, but there is certainly room for improvement when writing is concerned. The paper is well organized though and the notation (used mostly in Section 3) is clear and consistent.

Quality:
The design and justifications for the proposed LSTM-based-AUTO-Bi-LSTM for engine RUL prediction seem to be technically valid. While some of the observations made regarding the performance of LSTM-based-AUTO-Bi-LSTM hold, certain observations (particularly those made based on Figure 9) are not adequately discussed. Moreover, the performance of LSTM-based-AUTO-Bi-LSTM is demonstrated on only one of the four engine degradation datasets that the authors discussed in the paper, hence the effectiveness of LSTM-based-AUTO-Bi-LSTM is not well supported empirically (more details are included in the “Weaknesses” part of this review). Overall, the paper concerns an interesting and rather important prediction problem but is certainly not well developed and mature in terms of quality.

Novelty:
From a methodological perspective, the contribution of this work cannot be considered novel. In essence, the proposed architecture is solely based on a rather straightforward combination of an LSTM autoencoder and a supervised Bi-LSTM, both being widely used and well established neural network variants for sequence encoding.

Reproducibility:
The experiments were conducted on engine degradation data which are publicly available. On the other hand, the code for the proposed LSTM-based-AUTO-Bi-LSTM is not made available by the authors in the present anonymized version, however, one should be able to implement the architecture in a fairly straightforward manner by following Section 3.

**Details Of Ethics Concerns:**

Not applicable.

**Strength And Weaknesses:**

Strengths:

* The problem addressed in this paper is a problem of considerable importance as RUL is one of the most critical indicators of a component’s failure before the failure occurs. In a real-world setting, having reliable and accurate RUL estimates can substantially reduce maintenance costs and enforce proactive equipment replacement.

* To bypass manual feature engineering from the multivariate sensor datasets, the authors opt for well established neural network variants for sequence encoding that allow for automatic learning of latent features. This design choice suits their RUL task well as opposed to handcrafting features manually or using conventional machine learning methods to extract features directly from multivariate time series.

* The effectiveness of the proposed architecture has been assessed on engine degradation datasets simulated under various operational conditions, each consisting of multivariate time series representing records of several sensor channels that may characterize fault evolution.

* The authors provide a comprehensive and quite informative overview of LSTM autoencoders and supervised Bi-LSTM networks in Sections 3.1 and 3.2, respectively.

-------------------------------------------------------------------------------------------------

Weaknesses:

* In a couple of instances throughout the paper, the authors state that RUL prediction can be performed based on historical data or direct data extraction by adopting “model-based, data-driven, or hybrid methodologies”. Nevertheless, apart from data-driven methods, no model-based or hybrid methods have been included among the baselines. Namely, all of the considered baselines (summarized in Table A3 and Table A4) are indeed data-driven methods. Thus, I believe that this work would have benefited from a comparison of LSTM-based-AUTO-Bi-LSTM with several representative methods from the classes of model-based and hybrid methods, in addition to the already considered data-driven methods.

* The authors discuss that for the purpose of overcoming several challenges and improving RUL prediction accuracy, there has been “a prominent use of Deep Learning (DL) Methods, especially Recurrent Neural Networks (RNN) and Long Short-Term Memory (LSTM), besides other variations (Zhu et al., 2019; Li et al., 2020; Liu et al., 2020b)”. If that is the case, I am wondering why the proposed architecture has not been compared to some recently published RNN/LSTM-based variants specifically designed for RUL prediction, some of which are listed as follows:

  * Hong, C. W., Lee, K., Ko, M. S., Kim, J. K., Oh, K., & Hur, K. (2020, February). Multivariate time series forecasting for remaining useful life of turbofan engine using deep-stacked neural network and correlation analysis. In 2020 IEEE International Conference on Big Data and Smart Computing (BigComp) (pp. 63-70). IEEE.

  * Liu, H., Liu, Z., Jia, W., & Lin, X. (2020). Remaining useful life prediction using a novel feature-attention-based end-to-end approach. IEEE Transactions on Industrial Informatics, 17(2), 1197-1207.

  * Yu, W., Kim, I. Y., & Mechefske, C. (2020). An improved similarity-based prognostic algorithm for RUL estimation using an RNN autoencoder scheme. Reliability Engineering & System Safety, 199, 106926.

  * Yu, W., Kim, I. Y., & Mechefske, C. (2019). Remaining useful life estimation using a bidirectional recurrent neural network based autoencoder scheme. Mechanical Systems and Signal Processing, 129, 764-780.

  * Park, D., Kim, S., An, Y., & Jung, J. Y. (2018). LiReD: A light-weight real-time fault detection system for edge computing using LSTM recurrent neural networks. Sensors, 18(7), 2110.

  * Gugulothu, N., Tv, V., Malhotra, P., Vig, L., Agarwal, P., & Shroff, G. (2017). Predicting remaining useful life using time series embeddings based on recurrent neural networks. arXiv preprint arXiv:1709.01073.

  Although the proposed architecture is directly based on (Bi-)LSTMs, it has neither been compared to any of the above listed methods nor has any of their respective publications been cited and discussed as a part of the related work.

* The results from the comparison of LSTM-based-AUTO-Bi-LSTM with the state-of-the-art methods are included and discussed only in the case of the FD001 dataset. For some reason, the results obtained on the remaining three engine degradation datasets have been omitted from the paper as well as from the supplementary material / appendix. Therefore, I would encourage the authors to include the results obtained on these datasets and discuss them accordingly in the main paper, if possible. Alternatively, I would suggest that those results are included in the supplementary material / appendix.

* Figure 9 suggests that there seem to be several baselines that outperform LSTM-based-AUTO-Bi-LSTM in terms of RMSE. For instance, among the automatic feature learning methods, Bi-LSTM+CNN outperforms the proposed architecture by a significant margin. Unless I have somehow misinterpreted the findings from Figure 9 (in which case I would strongly suggest that the authors correct me), I would encourage the authors to elaborate on this observation of mine in their response; since, if such an observation holds, that brings into question the effectiveness of the proposed architecture.

* I consider that assessing RUL prediction performance simply by measuring RMSE is not sufficient. In addition to RMSE, I would recommend that the authors consider evaluation metrics that capture the accuracy of the methods’ predictions relative to the variation within the time series. Such metrics include: Mean Relative Absolute Error (MRAE) which is sensitive to outliers, Relative RMSE, and coefficient of determination.

-------------------------------------------------------------------------------------------------

Other (minor) weaknesses:

* Hyperparameter choices and experimental settings such as the number of training epochs, batch size, optimizers, loss functions, and so forth, are specific to the training process and are already mentioned in Section 4.4.1, thus I see no reason for those details to be duplicated in Section 4.4.2.

**Summary Of The Paper:**

This work utilizes (1) a Long Short-Term Memory (LSTM) autoencoder to learn unsupervised latent representations of multivariate time series data, and (2) a supervised Bidirectional LSTM (Bi-LSTM) that is initialized with the representations generated in step (1) and tasked to predict a continuous-valued output. The resulting architecture, referred to as LSTM-based-AUTO-Bi-LSTM, has been applied to the task of Remaining Useful Life (RUL) prediction based on engine degradation data from four datasets simulated under various operational conditions; and its effectiveness has been assessed and compared against several neural architecture-based baselines.

**Summary Of The Review:**

This work addresses the problem of RUL prediction, which is rather interesting and impactful considering that RUL is one of the most critical indicators of component failure. Nevertheless, in my view, the contributions of this work cannot be considered novel from a methodological perspective. Moreover, certain experiments appear to be missing from the paper (although the datasets concerning those experiments are discussed by the authors), and the effectiveness of the proposed architecture is not well supported empirically; among other weak points (outlined in the “Weaknesses” part of this review). That being said, the weak points of this paper seem to outweigh its strengths. Therefore, I am not convinced that this work is a good fit for ICLR. Nevertheless, I am looking forward to the authors’ response and I would be willing to adjust my score in case I have misunderstood or misinterpreted certain aspects of the work.

---

### Decision · Program_Chairs · 2023-01-20

**Decision:**

Reject

**Justification For Why Not Higher Score:**

All the reviews agree that this is reject.

**Justification For Why Not Lower Score:**

N/A

**Metareview: Summary, Strengths And Weaknesses:**

The paper discusses the problem of predicting the Remaining Useful Life of engine components. By pointing out that traditional machine learning methods cannot extract features directly from time series, the authors propose a method based on deep networks. The method is investigated in an experimental study.

The paper concerns rather an application of representation learning than contributes to this field. It mainly describes a specific case study, which certainly is interesting, but ICLR is not the right place for this paper in its current form. Moreover, the reviewers have pointed out several other flaws of the paper. Also the authors have agreed with some of the critical remarks included in the reviews.